# Fiber Optic Particle Plasmon Resonance-Based Immunoassay Using a Novel Multi-Microchannel Biochip

**DOI:** 10.3390/s20113086

**Published:** 2020-05-29

**Authors:** Chang-Yue Chiang, Chien-Hsing Chen, Chien-Tsung Wang

**Affiliations:** 1Graduate School of Engineering Science and Technology, National Yunlin University of Science and Technology, Yunlin 64002, Taiwan; chiangcy@yuntech.edu.tw; 2Bachelor Program in Interdisciplinary Studies, National Yunlin University of Science and Technology, Yunlin 64002, Taiwan; 3Department of Biomechatronics Engineering, National Pingtung University of Science and Technology, Pingtung 91201, Taiwan; 4Department of Chemical and Materials Engineering, National Yunlin University of Science and Technology, Yunlin 64002, Taiwan; ctwang@yuntech.edu.tw

**Keywords:** multi-microchannel, biochip, optical fiber, particle plasmon resonance, streptavidin, DNP, anti-DNP

## Abstract

A novel multi-microchannel biochip fiber-optic particle plasmon resonance (FOPPR) sensor system for the simultaneous detection of multiple samples. The system integrates a novel photoelectric system, a lock-in module, and an all-in-one platform incorporating optical design and mechanical design together to improve system stability and the sensitivity of the FOPPR sensor. The multi-microchannel FOPPR biochip has been developed by constructing a multi-microchannel flow-cell composed of plastic material to monitor and analyze five samples simultaneously. The sensor system requires only 30 μL of sample for detection in each microchannel. Moreover, the total size of the multi-microchannel FOPPR sensor chip is merely 40 mm × 30 mm × 4 mm; thus, it is very compact and cost-effective. The analysis was based on calibration curves obtained from real-time sensor response data after injection of sucrose solution, streptavidin and anti-dinitrophenyl (anti-DNP) antibody of known concentrations over the chips. The results show that the multi-microchannel FOPPR sensor system not only has good reproducibility (coefficient of variation (CV) < 10%), but also excellent refractive index resolution (6.23 ± 0.10 × 10^−6^ refractive index unit (RIU)). The detection limits are 2.92 ± 0.28 × 10^−8^ g/mL (0.53 ± 0.01 nM) and 7.48 ± 0.40 × 10^−8^ g/mL (0.34 ± 0.002 nM) for streptavidin and anti-DNP antibody, respectively.

## 1. Introduction

Automated high-throughput multi-analyte detection has become widely applied in environmental [1], chemical [2], biological and clinical diagnosis in recent years, drawing greater interest in related researches [3,4,5]. Compared with parallel, single-analyte assays, multi-analyte detection is characterized by shorter analysis time, simplified analytical procedure, minimized sample volume and enhanced test efficiency and cost effectiveness. Multi-analyte sensors have been used for detecting multi-analyte protein biomarkers and hazardous toxics in environmental pollutants. Typical examples include surface plasmon resonance (SPR) sensor [6,7], particle plasmon resonance sensor [8,9], electrochemical immunosensors [10], radioisotope-based bioassays or quantum dots [11,12,13,14,15,16,17,18,19]. Nevertheless, in some analytical approaches, enzymes and fluorescent dyes are first labeled in order to generate a physically readable signal from the recognition event [14,19]. In general, the labeling procedures are time consuming, and require trained users and sophisticated/expensive experimental tools and techniques [20].

In other research [21], there was development of a novel fiber-optic particle plasmon resonance (FOPPR) platform for real-time measurements. Using simple fiber-optics such as a transducer, the FOPPR sensing system is highly sensitive, thus making it a very attractive technique. As PPR is extremely sensitive to the change in the local refractive index occurring at the nanoparticle surface, reporter molecules are used in the FOPPR system to monitor binding biomolecules on the nanoparticle surface and their subsequent affinity interactions in real-time. In previous studies, the PPR sensor was used to detect various physical and chemical parameters, such as refractive index of the environment, food safety monitoring and antibody-antigen conjugation [9,22,23,24,25,26,27,28,29,30,31,32]. The PPR sensor experimentally demonstrated high sensitivity, good reproducibility and excellent stability in the analysis of targets in the above studies. FOPPR sensing technique mainly utilizes the multiple total internal reflection (TIR) schemes and the evanescent wave to enhance the absorption by gold nanoparticles (AuNPs), as well as the signal-to-noise ratio. When light propagates in the fiber core via consecutive TIR, the PPRs of immobilized AuNPs are excited by the evanescent field at the fiber core surface, thus attenuating the light transmitted through the fiber by interaction with AuNPs, as shown in Figure 1a. The PPR is the collective electron oscillations of metal nanoparticles. When the incident photon frequency is resonant with the oscillation frequency of conductive electrons, the absorption and scattering of the electromagnetic radiation are thus enhanced [9,25,30,31,32]. Hence, the FOPPR sensor for real-time direct monitoring of molecular interactions is based on the localized evanescent field absorption by the AuNPs upon biomolecular interaction. Figure 1b shows the resulting red-shift and increased extinction (decreased transmission intensity) measured at the distal end of the optical fiber [31]. Thus, maintenance of the optical stability of the light source in the FOPPR sensor is critical. This work proposed a novel multi-microchannel biochip of FOPPR sensor with a novel platform system to minimize the effects of fixing optical module (light source and detectors) and mechanical module (chip holder) fluctuation on the light source stability, in order to lessen baseline drift and improve signal-to-noise ratio.

In this study, we present a low-cost, rapid, real-time multi-analyte fiber-optic PPR-based biosensor, which can be used for the label-free detection of chemical and biological samples. A multi-microchannel FOPPR sensor biochip has been newly designed by constructing a multi-microchannel (five channels) flow-cell made up of plastic material [poly(methyl methacrylate), PMMA] for simultaneous analysis and the monitoring of multiple samples. Moreover, the flow cell used here is very small, with a dimension of only 20 mm × 1 mm × 1 mm (only 30 μL of sample was used for detection in each microchannel), thereby making the sensor system very compact and cost-effective. The multi-microchannel FOPPR module uses a microchip, which is merely 40 mm × 30 mm × 4 mm. The multi-microchannel FOPPR lab with an immunosensor chip has been developed by taking advantage of the high sensitivity, stability and reproducibility of the conventional FOPPR sensor through the use of an advanced microfluidic system and high-quality sensing platform. The all-in-one multi-microchannel sensor platform proposed in this study can lessen the baseline drift for the injection in samples. Its integrated flow-system can be switched between the FOPPR sensor and a storage device at any time. The developed system is more efficient than the existing multi-analyte biosensing systems.

## 2. Materials and Methods

### 2.1. Reagents and Materials

All materials and reagents were used as received and follows the previously reported [9]. Hydrogen tetrachloroaurate (III) trihydrate (HAuCl_4_), biotin, anti-biotin, bovine serum albumin (BSA) and anti-dinitrophenyl (anti-DNP) were purchased from Sigma-Aldrich. Cystamine dihydrochloride was purchased from Acros. Dinitrophenyl-epsilon-aminocaproic acid (DNP, molecular mass of 220 kDa) was purchased from ICN. Cetyltrimethylammonium bromide (CTAB), N-hydroxy-succinimide (NHS), 1-ethyl-3-(3-dimethylaminopropyl) carbodiimide hydrochloride (EDC), 2-(4-(2-hydroxyethyl)-1-piperazinyl)-ethanesulfonic aid (HEPES), (3-mercaptopropyl)-trimethoxy-silane (MPTMS, 98%) and streptavidin (molecular mass of 55 kDa) were purchased from Fluka. Sodium borohydride (NaBH_4_) was purchased from Lancaster. All the aqueous solutions were prepared with purified water from the Milli-Q water purification system (Millipore). The specific resistance was 18.2 MΩ. Multimode plastic-clad silica fiber (model F-MBC, Newport), with core and cladding diameters of 400 and 430 μm, respectively, was used. Poly (methyl methacrylate) (PMMA) was used for manufacturing multi-microchannel microfluidic biochip. The multi-microchannel biochip was composed of two poly (methyl methacrylate) (PMMA) plates, a cover and a bottom plate, with dimensions of 40 mm (width) × 30 mm (length) × 4 mm (thickness) and fabricated by using a CO_2_ laser [33]. The bottom plate contained multi-microchannels (five microchannels) with a depth of 1000 µm and a width of 1000 µm. It could accommodate the optical fiber, which had a coating diameter of 730 µm. A sample holder was designed with many injection ports and outlet ports. The total length of the optical fiber was 8 cm, while the sensing zone (the unclad portion) in the middle segment was 2 cm. Teflon tubing was then attached to the chip at inlet and outlet channels, as shown in Figure 2.

### 2.2. Multi-Microchannel Biochip Biosensing System

The multi-microchannel biochip sensing system integrated with a photoelectric system, as shown in Figure 2, comprises a light source module that consists of several light-emitting diodes (LED, model IF-E93, Industrial Fiber-optics) with a central wavelength at 530 ± 25 nm and an LED driver circuit, a multi-mirochannel FOPPR biochip, a detection module consisting of several photodiodes (PD, S1336-18BK, Hamamatsu), a photoreceiver amplification circuit (PAC, photodiode current/voltage converter and amplifier) and a lock-in module (dynamic signal acquisition module USB-9234 and LabVIEW software, National Instrument). The driving signal is generated by the LED driver circuit, where the circuit provides 1 kHz transistor-transistor logic (TTL) voltage. The amplified voltage signal, which means real-time light intensity, is processed with a 24-bit analog to digital converter (ADC), and analyzed by the National Instruments LabVIEW software.

### 2.3. The Refractive Index Sensitivity Test by a Multi-Microchannel Biosensing System

Spherical gold nanoparticles (AuNPs) used in this study was synthesized by using a slight modification of the seed-mediated growth method [9,29,31]. The AuNPs immobilization of on the unclad portion of optical fibers was prepared according to our previous procedures with minor modifications [25,29]. As shown in Figure 3a, the unclad portion of the optical fibers was submerged into vials of 2% solution of MPTMS in toluene for 6 hours. Then, the unclad portion of the optical fibers was immersed in AuNPs solution (the absorbance at 517 nm of 1.0 absorbance unit (A.U.)) for 4 h, so as to form a self-assembled colloidal monolayer of AuNPs on the surface of the partially unclad fiber. The fiber-optic spectrum system (Figure 3b) was then used to measure the extinction spectrum to examine the surface coverage distribution of the AuNPs on the FOPPR. As shown in Figure 3c, the coefficient of variation (CV) of maximum absorbance wavelength and maximum absorbance were 0.07% and 1.42%, respectively. The SEM image of the AuNPs on the optical fibers was inspected using an ultra-high resolution thermal field emission scanning electron microscope (JEOL JSM-7610F Plus, Tokyo, Japan) is shown in Figure 3d. To find out the transducing ability and refractive index sensitivity of the fiber-optic probe, sucrose solutions with various (weight percent) concentrations ranging from 6.8% to 41.7% corresponding to the refractive index (RI) with a range of 1.343–1.403 were prepared [31,34]. The sensitivity of the multi-microchannel sensor is defined as the change in the normalized intensity while measuring a particular concentration of analyte. Five or more replicates were performed for each measurement condition.

### 2.4. Functionalization of Fiber-Optic Probe

The fiber-optic probe was biofunctionalized according the previously suggested procedures [9,25,29,31]. As shown in Figure 4, AuNPs on the fiber-optic probes were modified due to the formation of a self-assembled monolayer of cystamine by immersing them in an aqueous solution of cystamine dihydrochloride (0.02 M) in deionized water for 2 h at room temperature, then rinsing them with deionized water. To functionalize biotin or DNP, the cystamine-modified probe was immersed separately in HEPES (containing 0.1 M of EDC, 0.025 M of NHS and 5 mM of biotin or 0.34 mM of DNP) for 2 h at room temperature. Finally, the fibers were rinsed thoroughly with PBS solution (pH 7.4, containing 150 mM of NaCl, 4 mM of KCl, 8.1 mM of Na_2_HPO_4_ and 1.47 mM of KH_2_PO_4_).

### 2.5. Preparation of Standard Biological Samples and Statistical Analysis

Stock standard solutions were prepared in PBS, with concentration level of 100 g/mL and stored in a freezer at −20 °C. The standard solution of ant-DNP and streptavidin was prepared in a concentration range of 1 × 10^−10^ to 1 × 10^−5^ g/mL by diluting the stock solution and storing at 4 °C until used. Matlab 2015b was used for statistical analysis.

## 3. Results and Discussion

### 3.1. Sensing Principle of the FOPPR

FOPPR sensing uses an incident light of narrow wavelength band to excite PPR. The intensity of light that exited the fiber was monitored. Molecular binding of analyte on the receptor-conjugated AuNP is transduced to locally increase the refractive index of the medium surrounding the AuNPs. As a result, the plasmon absorbance of the AuNPs was increased. Previous studies reported that the transmitted optical intensity decreases of AuNPs comprises of the integrand absorption of AuNPs, non-interacting plasmon resonance band due to isolated AuNPs, and coupled plasmon resonance band of interacting AuNPs [9,31,32]. In other words, when the plasmon absorbance of the AuNPs on the optical fiber increases upon molecular binding, the intensity of light exiting from the fiber decreases and follows the molecular binding event in real time [21,25].

In addition, the sensing system utilizes the properties of multiple total internal reflection and the evanescent wave formation in optical fiber to increase the absorption by AuNPs and signal-to-noise ratio. From the real-time data, the quantitative analysis of evanescent wave absorption measurements of the FOPPR sensor was conducted. The collected optical signal of a sensor immersed in an analyte solution (I_S_) was compared to the intensity of the sensor immersed in a blank solution (I_R_). The sensor response is defined as (I_R_ − I_S_)/I_R_ = ΔI/I_R_ [25,31].

### 3.2. Stability and Reproducibility of FOPPR by Multi-Microchannel Biochip

In order to improve the sensitivity, accuracy and reproducibility of the FOPPR sensor, lowering the noise and baseline drift of the system were the key challenges faced while conducting the experiment. In view of the above-mentioned problems related to the prior FOPPR sensor, this study integrated the optical design (light source and detectors) and mechanical design (chip holder) platform all-in-one to enhance system stability. The real-time signals with both the old and new FOPPR sensors were monitored in real-time for 8 h. Throughout the experimental procedures, the air conditioner was turned on and the temperature was set at 25 °C. The baseline drift with the new FOPPR sensor typically was under 0.013% every 8 h, while the baseline drift with the old FOPPR sensor varied from case to case and fell within the range of 0.165–3.1% every 8 h. The improvement reduced noise and baseline drift. Moreover, the electronics module after integration is very small with a dimension of only 8 cm × 10 cm; therefore, the total size of the FOPPR sensing system comprising the microchip is very compact and cost-effective.

Furthermore, to examine the light source power stability of the multi-microchannel biochip FOPPR system, a sensor chip was filled with a PBS solution. In our previous study, an estimate of the power stability of the sensor’s light source was 0.0091% per 1000 s [25]. In this study, with a further improvement in the power stability of the LED output by the new platform, optical design, and mechanical platform design, a power stability of 0.00045% per 1000 s was achieved. The temperature fluctuations are shown in Figure 5, where there is a negligible influence of the ambient temperature fluctuation on the light intensity of the sensor.

The FOPPR multi-microchannel (five channels) sensor was examined for detecting simultaneous multi-analysis of sucrose solutions with various refractive indices (1.343–1.403 RIU). Figure 6a shows the time course of each sensor response over the 1.343–1.403 refractive index range by injecting sucrose solutions into the multi-microchannel sensor chip. As seen, the sensor response had a sharp decrease to reach a steady state. A decreased response signal was observed in each sensing fiber when sucrose solutions of increasing RI were injected into the five channels. This was due to the increased evanescent field absorption by the immobilized AuNPs at higher surrounding RI. At the end of the injection cycle using deionized water, the response signal remained the same as the original signal at the beginning of the injection cycle, demonstrating that a stable immobilization of gold nanoparticle monolayer was achieved on each sensing fiber. As shown in the plots of I/I_0_ vs. RI, the correlation coefficients of these regressions are higher than 0.997 (Figure 6b) and the RI sensitivity is lower than 6.23 ± 0.10 × 10^−6^ RIU for all fiber sensors (Table 1), confirming the validity of this type of multi-microchannel sensing approach. The simultaneous multiple analyses using multi-microchannel sensors can reduce the analysis time. Furthermore, compared with the results from measuring transmission through a self-assembled monolayer of colloidal gold on glass or from the plasmonic coupling modes theoretical simulations, the refractive index resolutions of the multi-microchannel sensing approach are significantly larger [29,35,36]. A series of five repetitive measurements of sucrose solutions of various RIs were used to estimate the precision of the presented sensing platform. The coefficient of variation (CV) was 1.25–5.65%, indicating that the present FOPPR sensor offers excellent reproducibility.

### 3.3. Bio Selectivity—Nonspecific Adsorption Test

Prior to analyte detection, the non-specific adsorption tests were conducted to verify the results. Before injecting a sample solution, the DNP-functionalized FOPPR sensor was exposed to a PBS solution to check the system stability. Then, a solution of 1 × 10^−5^ g/mL of BSA and 1 × 10^−7^ g/mL of anti-DNP solution was injected into a sensor chip. The sensor response of DNP-functionalized FOPPR sensor in the presence of 1 × 10^−5^ g/mL of BSA solution was indistinguishable from the background signal, indicating negligible non-specific adsorption in relation to the concentration of BSA and no specific binding reaction with DNP was noticed. With anti-DNP, the response of signal showed a decrease (as shown in Figure 7), and thus, the temporal signal generated is presented as a molecular binding kinetic curve. Therefore, it is confirmed that the detected signal change was caused by the specific binding event present between the anti-DNP and the DNP on the sensing surface.

### 3.4. Detection of Receptor-Analyte Pairs by Multi-Microchannel FOPPR Lab on a Sensing Chip

The concentration-dependent signal changes were investigated to determine the sensitivity and reproducibility that can be achieved for the simultaneous detection of receptor-analyte pairs binding using this multi-microchannel (five channels) sensor. This study used a series of receptor-analyte pairs to examine the biosensing capability of the integrated systems, such as biotin/streptavidin and DNP/anti-DNP. The biotin was employed as a receptor which was functionalized on the cystamine-modified Au nanoparticle surface. The first step involved transformation of the carboxyl group on biotin into reactive N-hydroxysuccinimide ester by adding a mixture of EDC and NHS in the PBS buffer solution. The second step involved the reaction of the NH_2_ group present on cystamine with NHS ester to form an amide compound. Afterwards, PBS buffer solution was injected into each flow channel until a stable baseline result was observed; streptavidin solutions of various concentrations (5 × 10^−7^ to 5 × 10^−5^ g/mL) were then injected. Figure 8a shows the simultaneous temporal responses upon the sequential injection of streptavidin solutions with concentration ranging from 5 × 10^−7^ to 5 × 10^−5^ g/mL. It is noticed that the light intensity decreases with increasing streptavidin concentration. A steady decrease in FOPPR sensor signal response, which follows a molecular binding curve, was observed in each sensing probe. At the end of the injection cycle, the FOPPR sensing signal response did not increase or decrease by further injection of a PBS buffer, indicating that stable binding of streptavidin was achieved in each probe by the molecular interactions. In the LOD of the multi-microchannel FOPPR biosensing system, which represents the lowest detectable concentration of an analyte in a sample, the real-time optical responses (intensity monitoring) can be determined (limit of detection, S/N ratio of 3). The results show that the multi-microchannel system has good reproducibility and sensitivity. The calibration curves exhibit rather good linearity (correlation coefficient is 0.9955 (see Table 2 and Table 3)), as shown in Figure 8b, and the LOD of each sensor is determined to be lower than 2.92 ± 0.28 × 10^−8^ g/mL (0.53 ± 0.01 nM).

To further demonstrate the capability of our multi-microchannel sensor (five channels) as a biosensor, we also tested the system with the DNP/anti-DNP pair. The DNP was employed as a receptor on each probe surface. After a PBS buffer was injected simultaneously into each channel until a stable baseline was observed, anti-DNP solutions of different concentrations (1 × 10^−7^ to 1 × 10^−5^ g/mL) were injected into each flow channel. Figure 9a shows the simultaneous temporal responses upon the sequential injection of anti-DNP solutions with concentrations ranging from 1 × 10^−7^ to 1 × 10^−5^ g/mL. The results showed that the multi-microchannel system has good reproducibility and sensitivity (see Table 4 and Table 5). The calibration curves demonstrated in Figure 9b show rather good linearity (correlation coefficient is 0.9988) and the LOD of each sensor is determined to be lower than 7.48 ± 0.40 × 10^−8^ g/mL (0.34 ± 0.002 nM). A comparison of the analytical performance of our multi-microchannel biosensing with the other immunological methods the detection limit of the present FOPPR sensor showed the same order of magnitude or two order of magnitude lower (16 nM for biotin–streptavidin complex antibody complex by label-free optical method and 0.50  nM for DNP–anti-DNP antibody by optical fiber sensor) [35,37].

## 4. Conclusions

In this study, we have successfully demonstrated a novel multi-microchannel biochip FOPPR sensor system, and applied it to detect multiple chemical and biochemical samples. The proposed system is easy to fabricate and cost-effective, as it only requires a simple optical setup, thus making it more advantageous than the conventional SPR and ELISA systems. The integration of the multi-microchannel sensor platform can lessen baseline drift for injection in samples, and improve signal-to-noise ratio. The experimental results of the proposed system demonstrated linear calibration curves, good reproducibility (CV < 10%), excellent sensitivity (refractive index resolution = 6.23 ± 0.01 × 10^−6^ RIU) and low detection limit in the analysis of biochemical samples (2.92 ± 0.28 × 10^−8^ g/mL (0.53 ± 0.01 nM), 7.48 ± 0.40 × 10^−8^ g/mL (0.34 ± 0.002 nM) for streptavidin and anti-DNP antibody, respectively).

Although the proposed system has demonstrated good applicability, it still has some drawbacks, First, the detection range can be improved to achieve oriented immobilization of bioreceptors on the sensor surface by using specific self-assembled monolayers. Second, the large sensing surface area should be enhanced, and more functional chemical groups (e.g., -COOH) can be achievable by adjusting new self-assembled monolayers (e.g., carboxymethylated dextran, multiwalled carbon nanotubes). Third, other self-assembled monolayers can be immobilized on the surface of gold nanoparticles for more efficient covalent immobilization of biomolecules, thus enhancing the detection range of FOPPR biosensors [7].

Our future work will focus on the validation of assay reliability on complex real samples. Furthermore, the improvements on the system will focus on structural simplicity, easy fabrication and simple optical setup, so that the sensors can be miniaturized. The FOPPR system is expected to be widely applied to other clinically and environmentally important biological molecules in the future.

## Figures and Tables

**Figure 1 sensors-20-03086-f001:**
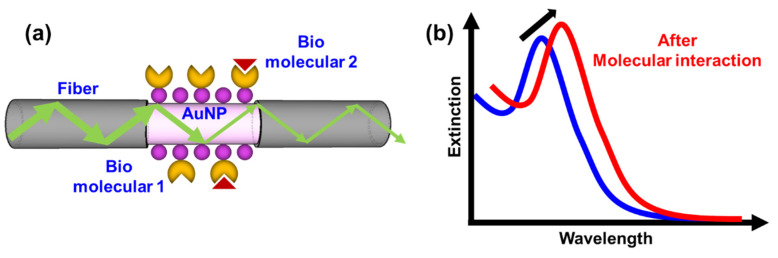
(**a**) Schematic diagram of the fiber-optic particle plasmon resonance (FOPPR) sensor; (**b**) Illustration of the FOPPR spectra with molecular interaction.

**Figure 2 sensors-20-03086-f002:**
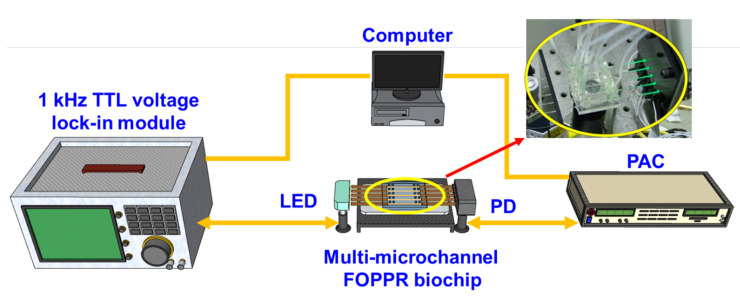
Schematic representation of the experimental setup of the multi-microchannel sensing system (Inset: The image of multi-microchannel FOPPR biochip). TTL: transistor-transistor logic; LED: light-emitting diodes; PD: photodiode; PAC: photoreceiver amplification circuit.

**Figure 3 sensors-20-03086-f003:**
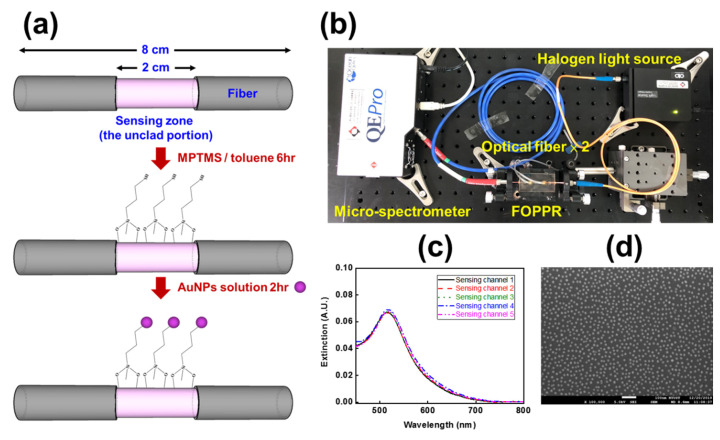
(**a**) Schematic of the seed-mediated growth method with AuNps on the unclad portion of optical fiber; (**b**) illustration of the fiber-optic spectrum system; (**c**) extinction spectra of the FOPPR sensors; (**d**) the SEM image of the gold nanoparticles (AuNPs) on the optical fiber.

**Figure 4 sensors-20-03086-f004:**
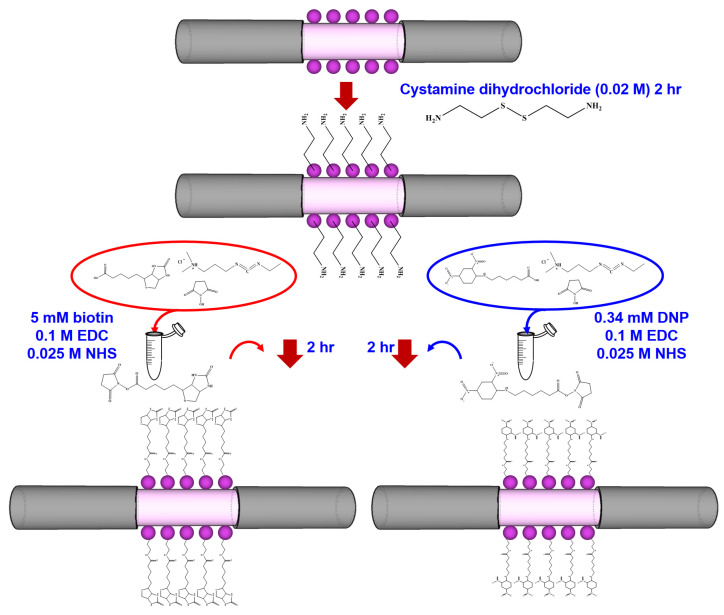
Schematic representation of the biofunctionalize setup of the optic probe. DNP: dinitrophenyl-epsilon-aminocaproic acid; EDC: 1-ethyl-3-(3-dimethylaminopropyl) carbodiimide hydrochloride; NHS: N-hydroxy-succinimide.

**Figure 5 sensors-20-03086-f005:**
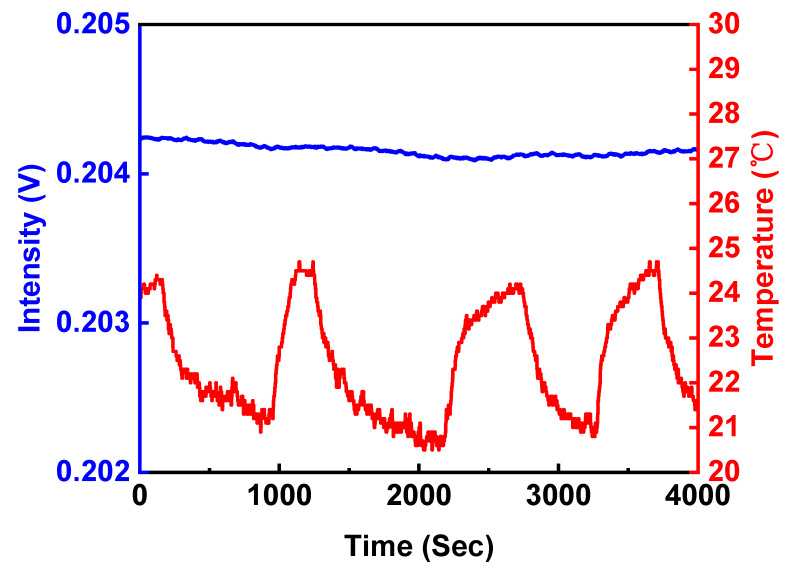
The temporal stability and effect of temperature fluctuation with the new generation of FOPPR sensor with proper heat-insulation.

**Figure 6 sensors-20-03086-f006:**
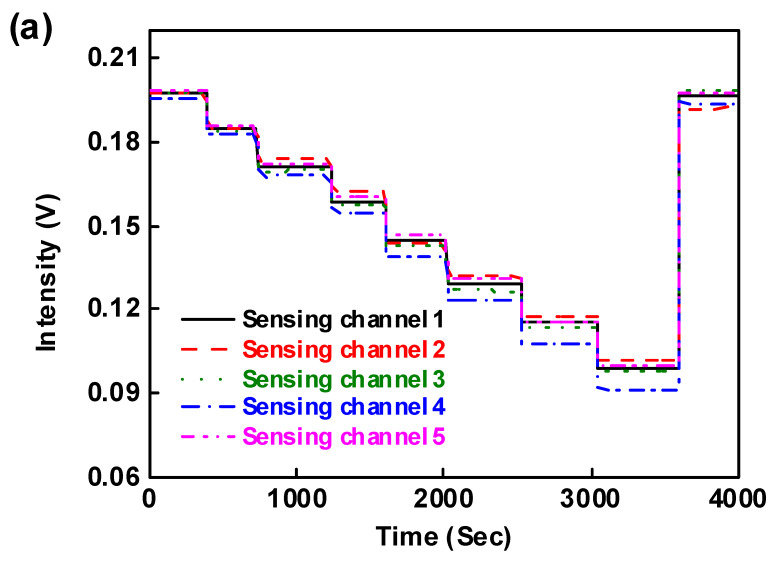
(**a**) Simultaneous temporal responses of all the sensing fibers present in the multi-microchannel FOPPR sensor after serial injection of samples with increasing refractive index; (**b**) calibration curves of sensor response vs. refractive index for the multi-microchannel sensing system.

**Figure 7 sensors-20-03086-f007:**
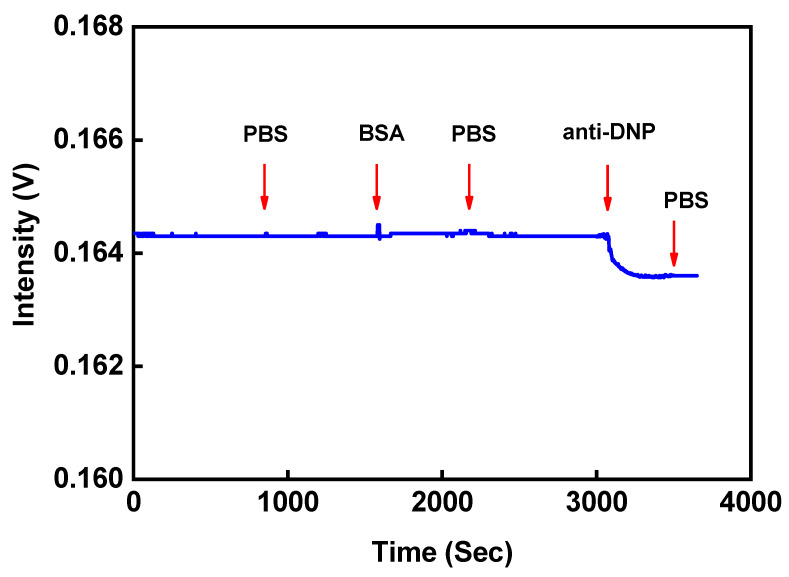
Nonspecific adsorption tests: Nonspecific adsorption tests. The sensorgram of DNP-functionalized FOPPR sensor in response to solutions.

**Figure 8 sensors-20-03086-f008:**
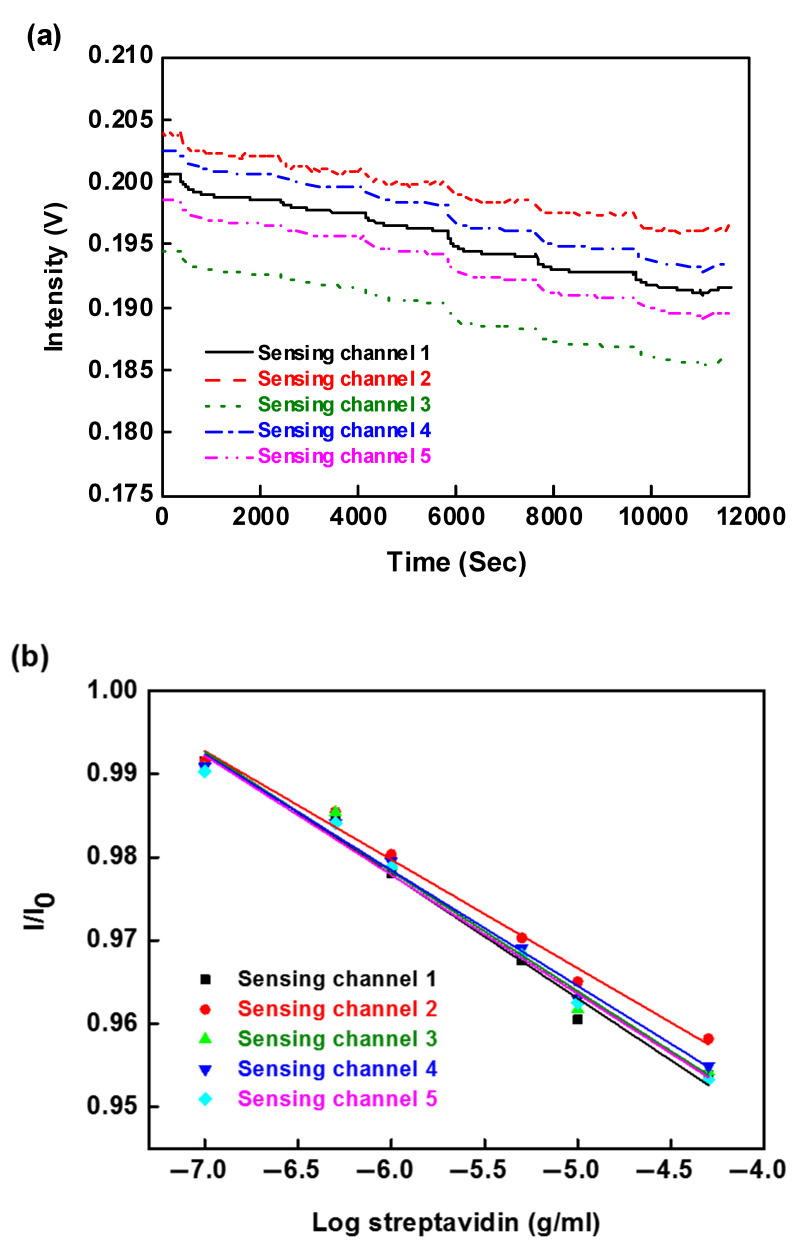
(**a**) Simultaneous temporal responses of all sensing fibers in the multi-microchannel FOPPR sensor with serial injection of streptavidin solutions; (**b**) calibration graph obtained with the biotin-functionalized probes for streptavidin by the multi-microchannel sensing system.

**Figure 9 sensors-20-03086-f009:**
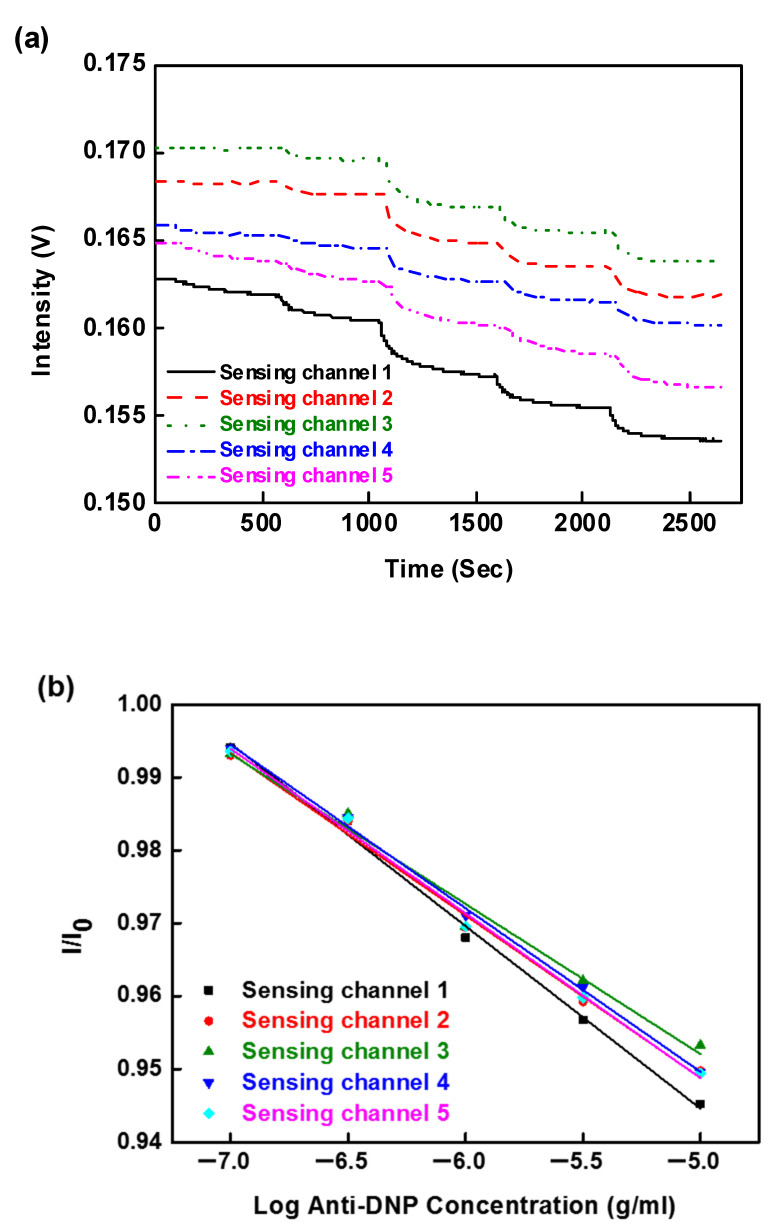
(**a**) Simultaneous temporal responses of all sensing fibers in the multi-microchannel FOPPR sensor with serial injection of anti-DNP solutions; (**b**) calibration graph obtained with the DNP-functionalized probes for anti-DNP by the multi-microchannel sensing system.

**Table 1 sensors-20-03086-t001:** Performance of the multi-microchannel FOPPR system in the detection of sucrose solution with various refractive indices.

	Slope	Correlation Coefficient (R)	Sensor Resolution (RIU)
Sensing channel 1	−7.14	0.9993	6.30 × 10^−6^
Sensing channel 2	−7.14	0.9984	6.30 × 10^−6^
Sensing channel 3	−7.23	0.9996	6.22 × 10^−6^
Sensing channel 4	−7.41	0.9978	6.07 × 10^−6^
Sensing channel 5	−7.19	0.9990	6.26 × 10^−6^
Average			6.23 ± 0.10 × 10^−6^

**Table 2 sensors-20-03086-t002:** Performance of the multi-microchannel FOPPR sensor system for detection of streptavidin/biotin binding.

	Slope	Correlation Coefficient (R)	LOD (g/mL)
Sensing channel 1	−0.0149	0.9929	2.5 × 10^−8^
Sensing channel 2	−0.0130	0.9950	3.2 × 10^−8^
Sensing channel 3	−0.0144	0.9913	3.1 × 10^−8^
Sensing channel 4	−0.0139	0.9955	2.8 × 10^−8^
Sensing channel 5	−0.0143	0.9949	3.0 × 10^−8^
Average			2.92 ± 0.28 × 10^−8^

**Table 3 sensors-20-03086-t003:** The reproducibility of detecting streptavidin/biotin binding by the multi-microchannel FOPPR sensor system.

	ΔI_1/_I_0_	ΔI_2/_I_0_	ΔI_3/_I_0_	ΔI_4/_I_0_	ΔI_5/_I_0_	ΔI_6/_I_0_
Sensing channel 1	0.0085	0.0153	0.0219	0.0324	0.0395	0.0462
Sensing channel 2	0.0085	0.0146	0.0197	0.0297	0.0349	0.0418
Sensing channel 3	0.0094	0.0147	0.0211	0.0314	0.0383	0.0457
Sensing channel 4	0.0092	0.0159	0.0205	0.0309	0.0370	0.0451
Sensing channel 5	0.0097	0.0157	0.0212	0.0318	0.0374	0.0467
CV%	5.95	4.36	4.02	3.25	4.53	4.31

**Table 4 sensors-20-03086-t004:** Performance of the multi-microchannel FOPPR sensor system for detection of anti-DNP/DNP binding.

	Slope	Correlation Coefficient (R)	LOD (g/mL)
Sensing channel 1	−0.0250	0.9977	7.5 × 10^−8^
Sensing channel 2	−0.0223	0.9970	7.3 × 10^−8^
Sensing channel 3	−0.0206	0.9929	7.8 × 10^−8^
Sensing channel 4	−0.0224	0.9988	6.9 × 10^−8^
Sensing channel 5	−0.0226	0.9984	7.9 × 10^−8^
Average			7.48 ± 0.40 × 10^−8^

**Table 5 sensors-20-03086-t005:** The reproducibility of detecting anti-DNP/DNP binding by the multi-microchannel FOPPR sensor system.

	ΔI_1/_I_0_	ΔI_2/_I_0_	ΔI_3/_I_0_	ΔI_4/_I_0_	ΔI_5/_I_0_
Sensing channel 1	0.0059	0.016	0.0320	0.0433	0.0548
Sensing channel 2	0.0070	0.016	0.03058	0.0408	0.0503
Sensing channel 3	0.0067	0.0154	0.0304	0.0379	0.0468
Sensing channel 4	0.0061	0.0155	0.0291	0.0388	0.0504
Sensing channel 5	0.0064	0.0156	0.0305	0.0402	0.0503
CV%	7.12	2.65	3.59	5.14	5.64

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
