# Peer review of "Fiber Optic Particle Plasmon Resonance-Based Immunoassay Using a Novel Multi-Microchannel Biochip"

_sensors, 2020, doi:10.3390/s20113086_

Round 1

Reviewer 1 Report

The multichannel sensor uses five optical fibers fixed on an experimental sensing platform to detect substances with different refractive index by means of light intensity detection. Compared with the design of multi parameter measurement on the same optical fiber sensor probe, it is relatively simple and provides a design idea of multi parameter measurement. Subsequently, different sensitive films are fixed on different channels, thus reducing a multi parameter design problem to several single parameter measurement problems.

Modification comments

  1. In this paper, "iber optical particle" in line 46 and "carbodiimine Hydrochloride (EDC) " in line 82 has two spelling mistakes.

  1. In part 2.3 of this paper, there is no detailed introduction to the fixed steps of AuNPs and no specific references are given. In view of this, I hope to give a specific combination diagram and explain it.

  1. In Section 2.4, there is no detailed schematic introduction for the self-assembly process of cystamine dihydrochloride. The specific use is the covalent combination with AuNPs or the fixation of cystamine dihydrochloride layer by Vander Ed Ley. Subsequently, there is no detailed introduction to the combination of HEPES, anti DNP and other substances with sensors, for which I hope to give a specific combination principle diagram and explain it.

  1. According to figure 2, the effect of temperature fluctuation on foppr sensor is given. When is the temperature changed? If the whole curve is obtained at a temperature higher than or lower than room temperature, does the light intensity change at room temperature? Can you make a comparison with the figure in Figure 2?

  1. For the comparison of Fig. 3 (b) and Fig. 5 (a), the format of the title size of the ordinates is inconsistent. In Figure 2, figure 3 (a) and Figure 4, the units in the brackets of the abscissa title are inconsistent, so it is expected to unify the writing format and make modifications.

  1. The first line of each paragraph is indented more than two characters, so the format of each paragraph is modified.

Author Response

Dear Reviewer:

    We would like to thank you for the thoughtful critiques of our manuscript. We have taken your comments fully into account and revised the manuscript accordingly. Our point-by-point responses to your comments are appended to this letter, please see the attachment. List of changes made in the revised manuscript based on your comments. Your comments are in black Arial font while our responses are in blue Times Roman font. We have also highlighted the changes made in the revised manuscript (with changes in red Times Roman font). Thank you very much for your kind consideration of this paper.

Yours sincerely,

Chien-Hsing Chen

Reviewer 2 Report

In this paper, Chiang et al. report a lab-on-a-chip fiber-optic sensor based on surface plasmon resonance for detection of a few samples. The paper is well-written but I have few questions/concerns listed below before final verdict:

  • It seems the working principle of each fiber-optic SPR sensor is based on gold nanoparticle deposition on un-clad portion of fiber-optics, and then sensitivity of resonance location on the refractive index of medium/analyte. However, the authors didn't explain this mechanism in detail: I would suggest to add a diagram, describing the cross section view of actual fiber-optic + gold layer to understand the actual sensing element of the system.
  • From the current schematic, I can see there are 5 fiber-optic channels in this sensor. As authors mentioned they deposit gold particles on unclad sections of these sensors. I was wondering if this deposition was uniform across all 5 fibers or if there was any difference in terms of thickness of nanofilms during deposition process (this can be the main reason we have different sensitivities for each element).
  • The "on-chip" part of the paper needs more clarification: from what I can see only the sensing elements (fiber-optics and LED/PDs) are on a chip and the system need full-size equipment to function.
  • Multiple analyte detection: What is the main advantage of having 5 different channels when you have to inject samples manually into the channels? Why not having only 1 channel and try it with 5 different samples at different times? I ask this question from a commercial point of view, as having 5 channels means more cost and a more complicated setup....
  • How should one try to improve detection range of the sensor? Is it even possible?
  • Please compare the performance and cost of your proposed device to commercially available sensors in the market or other recently published papers. The proposed device does not seems cheap to me...

Author Response

(The authors gave the same response as above.)

Round 2

Reviewer 2 Report

The authors addressed most of my comments and provide new plots/information as requested. I have no further comments for this paper and I suggest direct acceptance.